IPPP/22/74
PSI-PR-22-31

# Lepton pair production at NNLO in QED with EW effects

Sophie Kollatzsch[a,b] and Yannick Ulrich[c]

[a] Institut für Kern- und Teilchenphysik, TU Dresden, DE-01069 Dresden, Germany
[b] Paul Scherrer Institut, CH-5232 Villigen PSI, Switzerland
[c] Institute for Particle Physics Phenomenology, University of Durham,
South Road, Durham DH1 3LE, United Kingdom

We present a fully differential calculation of lepton pair production, taking into account the dominant next-to-next-to-leading order QED corrections as well as next-to-leading order electroweak and polarisation effects. We include all lepton masses, hard photon emission, as well as non-perturbative hadronic corrections. The corresponding matrix elements are implemented in the Monte Carlo framework McMule. In order to obtain a numerically stable implementation, we extend next-to-soft stabilisation, a universal technique based on a next-to-leading-power expansion, to calculations with polarised leptons. As an example, we show results tailored to the Belle II detector with the current setup as well as a potential future configuration that includes polarised beams.

## 1 Introduction

Thanks to its high luminosity, Belle II is expected to produce about 45 billion $\tau\tau$ events over its lifetime [1], roughly fifty times more than Belle I [2] and a hundred times more than at BaBar [3]. This increase in statistics will allow for precision measurements of very rare Standard Model (SM) decays such as $\tau \to \nu\bar{\nu}\ell\ell'\ell'$ or $\tau \to \nu\bar{\nu}\ell\gamma$ as well as put bounds on charged-lepton-flavour-violating decays such as $\tau \to \ell\ell'\ell'$. For the SM decays, even differential measurements in terms of Michel parameters will be possible. In this case, spin-spin correlations between the two taus of $ee \to \tau\tau$ can be exploited [4]. Further, it was recently proposed to measure the anomalous magnetic moment of the tau to $10^{-6}$ by exploiting transverse and longitudinal asymmetries [5].

Hence, the production cross section for $ee \to \tau\tau$ needs to be known as precisely as possible. For the centre-of-mass (CMS) energy used at Belle II ($\sqrt{s} \approx 10.5\,\text{GeV}$), QED effects still dominate even though electroweak (EW) effects start to become relevant. With EW effects, we mean all contributions due to EW interactions but without contributions due to pure QED. The Monte Carlo code KKMC [6–8] combines a parton shower with fixed-order EW corrections at next-to-leading order (NLO). It has been very useful for many experimental studies [1]. The EW effects were further studied with the SANC program, accounting for the polarisation of the incoming leptons [9].

However, the improvements expected from Belle II warrant a renewed theoretical effort. Currently no NNLO-QED calculation for $ee \to \ell\ell$ (with $\ell \neq e$) exists as the necessary two-loop integrals are not yet known with full mass dependence. However, a recent theoretical interest in $e$-$\mu$ scattering [10] was inspired by the MUonE experiment [11–13]. As part of this effort, the necessary integrals were computed in the limit of vanishing electron mass $m_e \to 0$ [14, 15]. This was very recently used to assemble to the full two-loop matrix element (squared) for $ee \to \ell\ell$ with $m_e = 0$ [16] which is an important part of the full NNLO-QED. Assuming $m_e$ is small compared to all other scales of the process, this matrix element can be used to obtain the full matrix element up to terms suppressed by $\mathcal{O}(m^2/Q^2)$. This *massification* procedure was first developed in [17–19] and later extended [20] to the case of a second, heavy mass.

However, the smallness of the electron mass means, that for a first estimate of the NNLO-QED correction, it is sufficient to just consider the electronic corrections, i.e. those due to the electron, and ignoring the more complicated mixed contributions. This can be done in a gauge-invariant manner by assigning different charges for each lepton family and only take contributions proportional to $Q_\ell^2 Q_e^6$. This was demonstrated at

NLO-QED [21] and then exploited to calculate the dominant NNLO-QED contributions to $e\mu \to e\mu$ [22,23]. Note that these $Q_\ell^2 Q_e^6$ corrections can be calculated exactly in the electron mass $m_e$ without approximation as the virtual corrections are just the heavy-quark form factor [24].

In this paper, we use the MCMULE framework to extend our previous calculation [22] of $e\mu \to e\mu$ to cover the electronic, or initial-state radiation (ISR), NNLO-QED corrections to $ee \to \ell\ell$. This means that our calculation includes the full NLO-QED (incl. the $Q_\ell^3 Q_e^3$ box contributions) but only the leading, i.e. $Q_\ell^2 Q_e^6$, NNLO-QED corrections. In particular, we do not include final-state radiation (FSR, $Q_\ell^6 Q_e^2$) and initial-final interference (IFI, $Q_\ell^3 Q_e^5$, $Q_\ell^4 Q_e^4$, and $Q_\ell^5 Q_e^3$) since these are suppressed.[1] We further treat the incoming electrons as polarised and include EW effects at NLO since these are becoming relevant at the energy at which Belle II operates.

Since the CMS energy of Belle II ($\sqrt{s} \approx 10.5\,\text{GeV}$) is significantly higher than for MUonE ($\sqrt{s} \approx 0.4\,\text{GeV}$), new numerical problems arise in the real-virtual matrix element, especially in the case of soft emission. These can be efficiently handled using next-to-soft (NTS) stabilisation [26], i.e. using a next-to-leading power (NLP) expansion if the emitted photon becomes soft.

This paper is organised as follows: in Section 2, we briefly summarise the calculation as implemented in the MCMULE framework. Next, we explain how NTS stabilisation changes when considering polarised particles in Section 3. Finally, we present some results for tau production cross sections and asymmetries, both in general and tailored to Belle II in Section 4 before concluding in Section 5.

## 2 Overview of the calculation

We consider the scattering process

$$e^+(p_1)e^-(p_2) \to Z/\gamma \to \tau^+(p_3)\tau^-(p_4)\{\gamma(p_5)\gamma(p_6)\}, \tag{1}$$

taking into account the full NLO-EW corrections and the electronic NNLO-QED corrections ($Q_\tau^2 Q_e^6$) but drop all remaining NNLO-QED terms, i.e. FSR ($Q_\tau^6 Q_e^2$) and IFI ($Q_\tau^3 Q_e^5$, $Q_\tau^4 Q_e^4$, and $Q_\tau^5 Q_e^3$) as discussed before. Since we are well below the $Z$ peak, we can expand the NLO-EW corrections by considering the masses of the EW bosons ($M_Z^2$, $M_W^2$, and $M_H^2$) much larger than all other scales of the process ($m_e^2$, $m_\tau^2$, $s = (p_1 + p_2)^2$, and $t = (p_1 - p_3)^2$). We then expand in the ratio of the heavy scale to the light scale, taking the first two terms of the expansion, i.e. keeping all terms $\mathcal{O}\big(\{s, t, m_e^2, m_\tau^2\}/\{M_Z^2, M_W^2, M_H^2\}\big)$. For simplicity, we write this as an expansion in $1/M_Z$. This procedure corresponds to how one expands in an effective field theory in $1/\Lambda$ while taking into account all effects up to and including dimension six. In this view, the base theory is QED and the underlying theory is not New Physics but rather the full SM. The resulting theory is a subset of what is often referred to as low-energy effective field theory (LEFT) [27–29].

Considering only the electronic, i.e. $Q_\tau^2 Q_e^6$, contributions at NNLO-QED means that we have exactly calculated the main source of logarithms in the electron mass, i.e. those terms containing $\alpha^2 \log^2(m_e^2/Q^2)$ (where $Q^2 \gg m_e^2$ is some other scale). Since we perform this calculation with full $m_e$ dependence we do include also power-suppressed terms that are $\propto Q_\tau^2 Q_e^6$. However, since such terms are also contained in the mixed IFI corrections, we have not included all logarithms and power-suppressed terms involving $m_e$. Similar logarithms in the tau mass do appear and those $\propto Q_\tau^6 Q_e^2$ (FSR) could be trivially calculated. However, since $m_e^2 \ll m_\tau^2 \sim Q^2$ these logarithms are not expected to be overly large.

Diagrams were generated with FeynArts [30] and QGraf [31] and calculated using Package-X [32] with full electron and tau mass dependence. Ultraviolet (UV) and infrared (IR) divergences are regularised in $d = 4 - 2\epsilon$ dimensions and the renormalisation is performed in the on-shell scheme up to NLO-EW and NNLO-QED. For the EW corrections, this means that we would like to use $e$, $M_W$, $M_Z$, $M_H$, and $m_\ell$ as

---

[1]While this paper was under review, the full NNLO-QED corrections for $e\mu \to e\mu$ were calculated [25] by using the two-loop matrix element with $m_e = 0$ [16] with massification. This calculation does indeed show that, at least for certain observables, the hierarchy of the different contributions holds with the $Q_\ell^6 Q_e^2$ remaining dominant. Extending [25] to $ee \to \ell\ell$ is planned for a future paper.

input parameters [33,34]. However, it is beneficial to use $G_F$ instead of $M_W$ as it has much higher precision

$$
\begin{aligned}
\alpha &= \frac{e^2}{4\pi} = \frac{1}{137.035999084}\,, \\
G_F &= 1.1663787 \times 10^{-11}\,\text{MeV}^{-2}\,, \\
M_Z &= 91\,187.6\,\text{MeV}\,, \\
M_H &= 125\,000\,\text{MeV}\,, \\
m_e &= 0.510998950\,\text{MeV}\,, \\
m_\tau &= 1\,776.86\,\text{MeV}\,.
\end{aligned}
\tag{2}
$$

For technical reason, the matrix elements in McMule are expressed through $s_W^2$ rather than $G_F$. $s_W^2 = 0.226202$ was obtained through

$$
M_W^2 s_W^2 = M_W^2 \left(1 - \frac{M_W^2}{M_Z^2}\right) = \frac{\alpha}{\sqrt{2}G_F}(1 + \Delta r),
\tag{3}
$$

where the one-loop expression of $\Delta r$ was taken from [34] to all orders in $1/M_Z$ but without including leading higher-order contributions.

The calculation is split into fermionic contributions that are due to fermionic vacuum polarisation (VP) effects (Section 2.1) and bosonic ones (Section 2.2).

Once properly renormalised, all matrix elements were implemented in version v0.4.0 of the publicly available parton-level integrator McMule [22, 35, 36]

https://mule-tools.gitlab.io

It performs the phase-space integration using the FKS$^\ell$ subtraction scheme [37], an all-orders QED extension of the FKS scheme [38, 39]. This allows us to calculate any IR-safe observable in a fully differential way.

Our calculation is performed with longitudinally polarised electrons (see for example [40, 41]). We introduce a polarisation vector $n_i$ along the beam direction for each particle that takes the form

$$
\begin{aligned}
n_i &= \big(0, 0, 0, P_i\big) \\
p_i &= \big(m_i, 0, 0, 0\big)
\end{aligned}
\tag{4}
$$

in the particle's rest frame. Of course $n_i \cdot p_i = 0$ in any frame. $|P_i| \le 1$ is the degree of polarisation that can be chosen as required by the beam parameters. To implement this, we modify the completeness relation of the spinors to

$$
u(p_i)\bar{u}(p_i) = (\slashed{p}_i + m_i)(1 + \slashed{n}_i \gamma_5)\,.
\tag{5}
$$

Alternatively, it may be simpler to calculate the matrix element for fully polarised initial states, i.e. with $P_i \equiv s_i = \pm 1$

$$
\mathcal{M}^{s_1 s_2} = \big|\mathcal{A}\big|^2_{P_1 = s_1, P_2 = s_2}\,.
\tag{6}
$$

This is for example done in OpenLoops [42, 43] which we will be using in Section 2.2. However, for most phenomenological applications we are interested in partial polarisation. For the (parity conserving) QED part, we can recover the general result as

$$
\mathcal{M} = \frac{1 + P_1 P_2}{2}\mathcal{M}^{\pm\pm} + \frac{1 - P_1 P_2}{2}\mathcal{M}^{\pm\mp}\,,
\tag{7}
$$

where $-1 \le P_i \le +1$ can be arbitrary.

## 2.1 Fermionic corrections

Up to NNLO, all fermionic corrections to our process are due to closed fermion bubbles. They can be split into leptonic contributions (electrons, muons, and taus) and non-perturbative hadronic effects (HVP).[2] At

---

[2]One technically also has to include top loops which are very suppressed at the energy scales we consider. Since the HVP contributions are ultimately extracted from data (see below), they have an inherit uncertainty which is roughly at the percent level – far bigger than top loops. Hence, the top quark is neglected in this calculation.

one-loop, these can be written in terms of the photonic and $Z$ currents for the lepton flavour $\ell = e, \tau$

$$j_\mu^{(\ell, \gamma)} = e\, \bar{v}(p, m_\ell)\gamma_\mu u(p', m_\ell)\,, \tag{8a}$$

$$j_\mu^{(\ell, Z)} = e\, \bar{v}(p, m_\ell)\gamma_\mu\Big(g_- P_L + g_+ P_R\Big)u(p', m_\ell)\,, \tag{8b}$$

with the momenta properly chosen. The $(Z\ell\bar{\ell})$-couplings are flavour-universal

$$g_- = -\frac{\frac{1}{2} - s_W^2}{s_W c_W} \quad \text{and} \quad g_+ = \frac{s_W}{c_W}\,. \tag{9}$$

The (renormalised) one-loop amplitude for the fermionic vacuum polarisation can be divided into three parts

$$
\begin{aligned}
\mathcal{A}_{\mathrm{vp}, f}^{(1)} ={}& \langle\!\!\!\!\text{---}\!\!\!\!\text{~}\Big(\Sigma_{\gamma\gamma, f}^{\mathrm{renorm.}}\Big)\!\!\!\!\text{~}\text{---}\!\!\!\!\rangle + \langle\text{---}\Big(\Sigma_{\gamma Z, f}^{\mathrm{renorm.}}\Big)\!\!\!\!\text{~}\text{---}\!\!\!\!\rangle + \langle\!\!\!\!\text{~}\text{---}\Big(\Sigma_{\gamma Z, f}^{\mathrm{renorm.}}\Big)\text{---}\!\!\!\!\rangle \\
&+ \langle\text{---}\Big(\Sigma_{ZZ, f}^{\mathrm{renorm.}}\Big)\text{---}\!\!\!\!\rangle + \mathcal{O}\Big(\frac{1}{M_Z^4}\Big) \\[2mm]
={}& \mathcal{A}_{\gamma\gamma, \mathrm{vp}, f}^{(1)} + \mathcal{A}_{\gamma Z, \mathrm{vp}, f}^{(1)} + \mathcal{A}_{ZZ, \mathrm{vp}, f}^{(1)} + \mathcal{O}\Big(\frac{1}{M_Z^4}\Big) \\[2mm]
={}& \frac{1}{s^2}j_\mu^{(e, \gamma)}j_\mu^{(\tau, \gamma)}\Sigma_{\gamma\gamma, f}^{\mathrm{renorm.}}(s) + \frac{1}{s(s - M_Z^2)}\Big(j_\mu^{(e, Z)}j_\mu^{(\tau, \gamma)} + j_\mu^{(e, \gamma)}j_\mu^{(\tau, Z)}\Big)\Sigma_{\gamma Z, f}^{\mathrm{renorm.}}(s) \\
&+ \frac{1}{(s - M_Z^2)^2}j_\mu^{(e, Z)}j_\mu^{(\tau, Z)}\Sigma_{ZZ, f}^{\mathrm{renorm.}}(s) + \mathcal{O}\Big(\frac{1}{M_Z^4}\Big)\,,
\end{aligned}
\tag{10}
$$

where the transversal fermionic self-energies $\Sigma_{ij, f}^{\mathrm{renorm.}}$ are renormalised in the on-shell scheme with the conditions [34][3]

$$\Sigma_{\gamma\gamma}^{\mathrm{renorm.}}(0) = 0\,, \quad \mathrm{Re}\Big[\Sigma_{ZZ}^{\mathrm{renorm.}}(M_Z^2)\Big] = 0\,, \quad \Sigma_{\gamma Z}^{\mathrm{renorm.}}(0) = 0\,, \quad \mathrm{Re}\Big[\Sigma_{\gamma Z}^{\mathrm{renorm.}}(M_Z^2)\Big] = 0\,. \tag{11}$$

Fermionic contributions due to other particles in the EW sector (such as the Higgs) are suppressed by at least $\mathcal{O}(1/M_Z^4)$ and hence already dropped. Corrections due to boson loops are included in Section 2.2.

For the $ZZ$ term, we extract the part that is $\mathcal{O}(1/M_Z^2)$ by defining a constant $C$

$$\mathcal{A}_{ZZ, \mathrm{vp}, f}^{(1)} = \frac{1}{M_Z^2}j_\mu^{(e, Z)}j_\mu^{(\tau, Z)}C + \mathcal{O}\Big(\frac{1}{M_Z^4}\Big) \tag{12}$$

that arises from the renormalisation of $M_Z$. Hence, it is determined through the fermionic part of the (unrenormalised) self-energy $\Sigma_{ZZ, f}(Q^2)$ at $Q^2 = M_Z^2$ where it can be calculated perturbatively as it has no kinematic dependence

$$C = \frac{\alpha}{6\pi}\sum_f \frac{(I_3^f)^2 - 2s_W^2 I_3^f Q_f + 2s_W^4 Q_f^2}{c_W^2 s_W^2}\,. \tag{13}$$

From the renormalisation conditions (11) we also find the explicit expressions for the renormalised self-energies [45]

$$
\begin{aligned}
\Sigma_{\gamma\gamma}^{\mathrm{renorm.}}(s) &= \Sigma_{\gamma\gamma}(s) - \Sigma_{\gamma\gamma}(0) = \Sigma_{\gamma\gamma}(s)\,, \\
\Sigma_{\gamma Z}^{\mathrm{renorm.}}(s) &= \Sigma_{\gamma Z}(s) - \Sigma_{\gamma Z}(0) - \frac{s}{M_Z^2}\Big(\mathrm{Re}\big[\Sigma_{\gamma Z}(M_Z^2)\big] - \Sigma_{\gamma Z}(0)\Big)\,.
\end{aligned}
\tag{14}
$$

---

[3]In the interest of clarity we keep the renormalisation condition for the photonic self-energy explicitly in (11) even though $\Sigma_{\gamma\gamma}(0) = 0$ already at the unrenormalised level. For the same reason we also retain $\Sigma_{ij, f}(0)$ in (15). Here, the fermionic part of the $\gamma Z$ term, $\Sigma_{\gamma Z, f}(0)$, vanishes as well [44].

By defining the fermionic VP function $\hat{\Pi}_{ij}$ [45, 46]

$$\Sigma_{ij,f}(Q^2) \equiv \Sigma_{ij,f}(0) + Q^2\,\hat{\Pi}_{ij}(Q^2)\,, \tag{15}$$

the $\gamma\gamma$ and $\gamma Z$ amplitudes can be written as

$$
\begin{aligned}
&\mathcal{A}^{(1)}_{\gamma\gamma,\mathrm{vp},f} + \mathcal{A}^{(1)}_{\gamma Z,\mathrm{vp},f} \\
&= \frac{1}{s} j^{(e,\gamma)}_\mu j^{(\tau,\gamma)}_\mu \hat{\Pi}_{\gamma\gamma}(s) + \frac{1}{(s-M_Z^2)}\Big( j^{(e,Z)}_\mu j^{(\tau,\gamma)}_\mu + j^{(e,\gamma)}_\mu j^{(\tau,Z)}_\mu \Big)\Big( \hat{\Pi}_{\gamma Z}(s) - \mathrm{Re}\Big[\hat{\Pi}_{\gamma Z}(M_Z^2)\Big]\Big).
\end{aligned}
\tag{16}
$$

Using the definitions of `alphaQED` [46]

$$\hat{\Pi}_{\gamma Z}(Q^2) = \frac{1}{s_W c_W}\Big( \hat{\Pi}_{3\gamma}(Q^2) - s_W^2 \hat{\Pi}_{\gamma\gamma}(Q^2)\Big), \tag{17}$$

where the 3 refers to the third component of the isospin current and $\gamma$ to the QED current, we arrive at the final expression

$$
\begin{aligned}
\mathcal{A}^{(1)}_{\mathrm{vp},f} =\; & \frac{1}{s} j^{(e,\gamma)}_\mu j^{(\tau,\gamma)}_\mu \hat{\Pi}_{\gamma\gamma}(s) \\
& - \frac{1}{M_Z^2}\Big( j^{(e,Z)}_\mu j^{(\tau,\gamma)}_\mu + j^{(e,\gamma)}_\mu j^{(\tau,Z)}_\mu \Big)\left( \frac{1}{s_W c_W}\hat{\Pi}_{3\gamma}(s) - \frac{s_W}{c_W}\hat{\Pi}_{\gamma\gamma}(s)\right) \\
& + \frac{1}{M_Z^2}\Big( j^{(e,Z)}_\mu j^{(\tau,\gamma)}_\mu + j^{(e,\gamma)}_\mu j^{(\tau,Z)}_\mu \Big)\left( \frac{1}{s_W c_W}\mathrm{Re}\Big[\hat{\Pi}_{3\gamma}(M_Z^2) - s_W^2 \hat{\Pi}_{\gamma\gamma}(M_Z^2)\Big]\right) \\
& + \frac{1}{M_Z^2} j^{(e,Z)}_\mu j^{(\tau,Z)}_\mu C + \mathcal{O}\Big(\frac{1}{M_Z^4}\Big).
\end{aligned}
\tag{18}
$$

Here, all non-perturbative $\hat{\Pi}_{ij}$ (including $\hat{\Pi}_{ij}(M_Z^2)$) are taken from `alphaQED` and can be obtained more or less directly from $R$ ratio data. However, $\hat{\Pi}_{3\gamma}$ is sensitive to fermions flavour in a different way than $\hat{\Pi}_{\gamma\gamma}$, necessitating a flavour recombination. The simplest strategy for this is assuming that $\mathrm{SU}(3)_f$ is an exact symmetry which results in $\hat{\Pi}_{3\gamma} = \frac{1}{2}\hat{\Pi}_{\gamma\gamma}$. `alphaQEDc19` uses a more complicated strategy based on vector meson dominance (VMD) [46]. The perturbative, leptonic contributions to $\hat{\Pi}_{ij}$ are included in the usual way by calculating the corresponding one-loop diagrams.

Beyond NLO, we have to account for QED-VP insertions into loop diagrams. This is done using the hyperspherical method [47, 48] that was used for $\mu$-$e$ scattering [49] and implemented in McMule for $\mu$-$e$, $\ell$-$p$, and Møller scattering [22, 50]. There are further two types of leptonic self-energy corrections at two loop in QED: the product of two one-loop self-energy bubbles ("bubble chain") and the genuine two-loop self-energy that can be obtained from [51]. Since these are fully perturbative, their inclusion is straightforward.

## 2.2   Bosonic corrections

At NLO, the bosonic corrections are given by two separately divergent types of contributions: real (R) and virtual (V).

The virtual contribution includes in total about 250 one-loop diagrams. The majority of them is due to self-energy corrections arising from purely bosonic loops such as $W$ corrections to the photon propagator (all NLO-EW). The remaining diagrams can be divided into vertex correction diagrams for the electronic $(Q_\tau^2 Q_e^4)$ as well as for the tauonic $(Q_\tau^4 Q_e^2)$ part and box contributions $(Q_\tau^3 Q_e^3)$ involving box diagrams with two photons (NLO-QED), one heavy boson ($Z, W, H$, or Goldstone bosons) and one photon, or two heavy bosons (all NLO-EW). The latter can still contribute at $\mathcal{O}(1/M_Z^2)$ even though two heavy boson propagators enter the calculation. Once the calculation and expansion of the one-loop diagrams is completed, we renormalise as discussed at the beginning of Section 2.

At NNLO-QED, we have three separately divergent types of contributions: virtual-virtual (VV), real-virtual (RV), and real-real (RR).

For the electronic corrections, the VV can be obtained from the on-shell renormalised heavy-quark form factor [24] which is written in terms of harmonic polylogarithms (HPLs) [52]. This allows for trivial analytic continuation into the time-like region we are interested in.

We use OpenLoops [42, 43] in its standard mode for the RV corrections. While OpenLoops is extremely stable, its standard mode may not be sufficient for soft or collinear emission, especially in the case of small fermion masses. To address this issue, we use NTS stabilisation [26]. The basic idea of this method is to switch to an expanded matrix element if the (rescaled) photon energy $\xi = 2E_\gamma/\sqrt{s}$ drops below a certain cut-off. This cut-off is usually varied between $10^{-5}$ and $10^{-2}$ to ensure that the final result does not depend on its value.

Below the cut-off, we use a matrix element that is expanded for small photon energies up to NLP. To do this, we use an extension of the LBK theorem [53, 54] to one loop [55, 56] that we will discuss in the next section.

# 3 LBK theorem for polarised particles

Following [55, 56], we will extend the LBK theorem to polarised cross sections at one loop. We use $\xi$ as the expansion parameter and write the photon momentum as $p_\gamma = \xi k$.

To better understand what happens at one loop, let us first review the changes in the case of polarised particles in the proof at tree-level where similar results have been obtained before [57, 58]. By splitting $\mathcal{A}^{(0)}_{n+1}$ into contributions from internal and external legs

$$
\mathcal{A}^{(0)}_{n+1} = \sum_i \left( \begin{array}{c} p_i \quad p_\gamma \\ \Gamma^{\text{ext}} \end{array} \right) + \begin{array}{c} p_\gamma \\ \Gamma^{\text{int}} \end{array} \,, \tag{19}
$$

and using gauge invariance, the NTS contribution can be written as [53–56]

$$
\mathcal{A}^{(0)}_{n+1} = \sum_i Q_i \left( \frac{1}{\xi} \frac{\epsilon \cdot p_i}{k \cdot p_i} \Gamma^{\text{ext}}(\{p\}) - \frac{\Gamma^{\text{ext}}(\{p\}) \slashed{k} \slashed{\epsilon}}{2k \cdot p_i} - \left[ \epsilon \cdot D_i \Gamma^{\text{ext}}(\{p\}) \right] \right) u(p_i) + \mathcal{O}(\xi^1) \,, \tag{20}
$$

with the LBK operator

$$
D_i^\mu = \frac{p_i^\mu}{k \cdot p_i} k \cdot \frac{\partial}{\partial p_i} - \frac{\partial}{\partial p_{i,\mu}} \,. \tag{21}
$$

When squaring $\mathcal{A}^{(0)}_{n+1}$, we have to consider the interference between the leading-power (LP, $\mathcal{O}(1/\xi)$ at amplitude-level) and NLP ($\mathcal{O}(\xi^0)$ at amplitude-level) terms. To do this in the unpolarised case, we would use the identity

$$
\frac{u(p_i)\bar{u}(p_i)\slashed{\epsilon}\slashed{k} + \slashed{k}\slashed{\epsilon}u(p_i)\bar{u}(p_i)}{2k \cdot p_i} = \frac{\epsilon \cdot p_i}{k \cdot p_i} \slashed{k} - \slashed{\epsilon} = \epsilon \cdot D_i \Big( u(p_i)\bar{u}(p_i) \Big) \tag{22}
$$

since $u(p_i)\bar{u}(p_i) = \slashed{p}_i + m_i$. In the polarised case, we have to use (5) and (22) gets modified accordingly

$$
\frac{u(p_i)\bar{u}(p_i)\slashed{\epsilon}\slashed{k} + \slashed{k}\slashed{\epsilon}u(p_i)\bar{u}(p_i)}{2k \cdot p_i} = \left[ \epsilon \cdot D_i - \frac{\epsilon_\mu k_\nu - \epsilon_\nu k_\mu}{k \cdot p_i} n_{i,\nu} \frac{\partial}{\partial n_{i,\mu}} \right] \Big( u(p_i)\bar{u}(p_i) \Big) \,. \tag{23}
$$

Hence, the matrix element is finally obtained after summing over the polarisation of the photon

$$
\mathcal{M}^{(0)}_{n+1}(\{p\}, k) = \sum_{ij} Q_i Q_j \Big( - \frac{1}{\xi^2} \frac{p_i \cdot p_j}{(k \cdot p_i)(k \cdot p_j)} + \frac{1}{\xi} \frac{p_j \cdot D_i}{k \cdot p_j} + \frac{1}{\xi} \frac{p_{j,\mu} k_\nu - p_{j,\nu} k_\mu}{(k \cdot p_i)(k \cdot p_j)} n_{i,\nu} \frac{\partial}{\partial n_{i,\mu}} \Big) \mathcal{M}^{(0)}_n(\{p\})
$$
$$
+ \mathcal{O}(\xi^0) \,. \tag{24}
$$

Thus, the calculation of the NTS term at tree-level remains straightforward as we just need to also calculate the derivatives w.r.t. the polarisation vector.

To extend this discussion to the one-loop level, we use the method of regions [59]. It was shown in [55], that the amplitude of the soft contribution is

$$
\mathcal{A}^{(1),\text{soft}}_{n+1} = - \sum_{i \neq j} Q_i^2 Q_j (i\mathcal{A}^{(0)}_n) \Big( \frac{p_i \cdot \epsilon}{k \cdot p_i} - \frac{p_j \cdot \epsilon}{k \cdot p_j} \Big) S(p_i, p_j, k) + \mathcal{O}(\xi^1) \,. \tag{25}
$$

The function $S(p_i, p_j, k) \sim k$ can be calculated universally and is presented in [55]. The hard contribution is closely related to the LBK theorem (24) with one important subtlety related to the following external leg corrections [56]

$$\mathcal{A}_{\text{ext},i}^{(1)} = p_i \underbrace{\phantom{xx}}_{}\; \overset{p_\gamma}{\text{(diagram)}} \Gamma^{\text{ext}} + p_i \; \overset{p_\gamma}{\text{(diagram)}} \Gamma^{\text{ext}} + p_i \; \overset{p_\gamma}{\text{(diagram)}} \delta m \; \Gamma^{\text{ext}} \,. \tag{26}$$

The vertex correction to the soft photon emission spoils the basic assumption of the LBK theorem that diagrams with internal emission do not contain any $1/p_\gamma$ poles. Further, the self-energy correction is technically an external correction and could be expanded using the normal LBK theorem. Hence, these contributions do not reduce to the non-radiative amplitude. Instead, one can show that (26) results in an extra contribution of the form

$$\mathcal{A}_{\text{ext},i}^{(1)} = Q_i^3 \Gamma^{\text{ext}} \epsilon \cdot H \, u(p_i) \tag{27}$$

$$H_\mu = \frac{1}{m_i} \gamma^\mu - \frac{p_i^\mu}{m_i (k \cdot p_i)} \slashed{k} - \frac{1}{k \cdot p_i} \gamma^\mu \slashed{k} \,. \tag{28}$$

When interfering this contribution with the LP term of (20) we find

$$\mathcal{M}_{\text{ext},i}^{(1)} = \frac{1}{\xi} \sum_j Q_j \frac{\epsilon^* \cdot p_j}{k \cdot p_j} \Gamma^{\text{ext}} \epsilon \cdot H \, u(p_i) \bar{u}(p_i) \Gamma^{\text{ext},\dagger} + \text{h.c.} \,. \tag{29}$$

In the unpolarised case, this contribution vanishes after some Dirac algebra. However, if $n_i \neq 0$, it does not and instead results in

$$\mathcal{M}_{\text{ext},i}^{(1)} = -\frac{1}{\xi} \sum_j \frac{Q_i^3 Q_j}{16\pi^2} \frac{1}{(k \cdot p_i)(k \cdot p_j)} \Big[ (2n \cdot p_j) k^\mu - (2n \cdot k) p_j^\mu \Big] \left[ \frac{\partial}{\partial n_{i,\mu}} \mathcal{M}_n^{(0)} - \frac{p_i^\mu}{m_i} \mathcal{M}_n^{(0),\prime} \right] \,. \tag{30}$$

Here, we need a modified version of the tree-level matrix element

$$\mathcal{M}_n^{(0),\prime} = \Gamma^{\text{ext}} (\slashed{p}_i + m_i) \gamma^5 \Gamma^{\text{ext},\dagger} \,. \tag{31}$$

We can now write down a version of the LBK theorem that is valid both at one-loop and in the case of polarised external particles

$$\begin{aligned}
\mathcal{M}_{n+1}^{(1)} = \sum_{i,j} Q_i Q_j \Bigg\{ &-\frac{1}{\xi^2} \frac{p_i \cdot p_j}{(p_i \cdot k)(p_j \cdot k)} \mathcal{M}_n^{(1)} + \frac{1}{\xi} \frac{p_j \cdot D_i[\mathcal{M}_n^{(1)}]}{k \cdot p_j} \\
&+ \frac{1}{\xi} \frac{p_j^\mu (k \cdot n_i) - k_\mu (p_j \cdot n_i)}{(p_i \cdot k)(p_j \cdot k)} \left[ \frac{\partial}{\partial n_{i,\mu}} \mathcal{M}_n^{(1)} + \frac{Q_i^2}{8\pi^2} \left( \frac{\partial}{\partial n_{i,\mu}} \mathcal{M}_n^{(0)} - \frac{p_i^\mu}{m_i} \mathcal{M}_n^{(0),\prime} \right) \right] \Bigg\} \\
&+ \frac{1}{\xi} \sum_{l,i \neq j} Q_i^2 Q_j Q_l \left[ \frac{p_i \cdot p_l}{(p_i \cdot k)(p_l \cdot k)} - \frac{p_l \cdot p_j}{(p_l \cdot k)(p_j \cdot k)} \right] \times 2\, \mathcal{S}(p_i, p_j, k) \mathcal{M}_n^{(0)} + \mathcal{O}(\xi^0) \,.
\end{aligned} \tag{32}$$

The new term $\mathcal{M}_n^{(0),\prime}$, while easy to calculate, has severe consequences for the structure of the NTS approximation. Every other term in (32) is *directly* related to the reduced process, either at one-loop ($\mathcal{M}_n^{(1)}$) or tree-level ($\mathcal{M}_n^{(0)}$). $\mathcal{M}_n^{(0),\prime}$, on the other hand, is a new structure that spoils the elegance of the LBK theorem and its extensions.

We have numerically verified that (32) is correct by taking the limit $\xi \to 0$ of the real-virtual matrix element relevant for this process (as shown in Figure 1) and also for $\mu \to \nu \bar{\nu} e \gamma$.

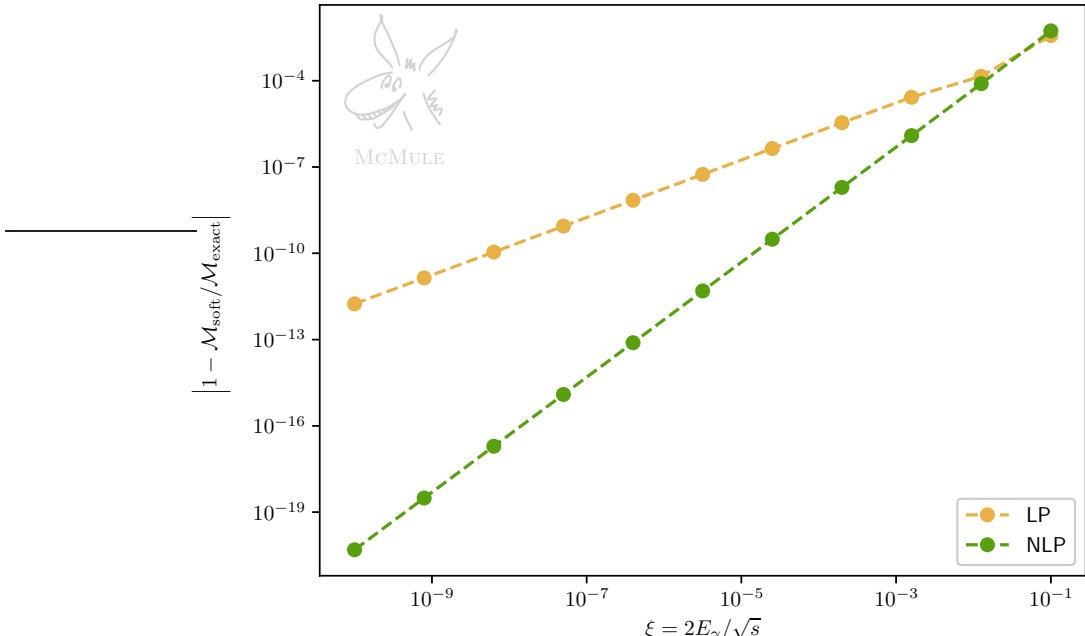

Figure 1: Convergence of the soft limit at LP and NLP of the dominant one-loop corrections to $e^+e^- \to \ell^+\ell^-\gamma$. The reference value $\mathcal{M}_{\text{exact}}$ is calculated with arbitrary precision in Mathematica.

## 4  Results

To validate our calculation, we have crossed it to $e\mu \to e\mu$ and compared the NLO-EW with [21] and the NNLO-QED with [22, 23] at the level of the differential distributions. We found full agreement in both cases.

In the following, we present some results for $e^+e^- \to \tau^+\tau^-$ at $\sqrt{s} = 10.5830052\,\text{GeV}$ both without any cuts and tailored to Belle II. We stress that these are just examples and that McMule can calculate any IR-safe observable.

We write the total cross section as

$$\sigma = \sigma_{\text{QED}} + \sigma_{\text{EW}} = \sigma_{\text{QED}}^{(0)} + \sigma_{\text{QED}}^{(1)} + \sigma_{\text{QED}}^{(2)} + \sigma_{\text{EW}}\,, \tag{33}$$

which is divided into the pure QED and the EW part. The former are further split into LO ($\sigma_{\text{QED}}^{(0)}$), NLO ($\sigma_{\text{QED}}^{(1)}$), and NNLO ($\sigma_{\text{QED}}^{(2)}$) contributions. Note that we do not split $\sigma_{\text{EW}} \equiv \sigma_{\text{EW}}^{(0)} + \sigma_{\text{EW}}^{(1)}$ as they turn out to be similar in size. As explained in the next section, this is the result of a cancellation within the LO-EW contributions.

In the interest of Open Science, all raw data, analysis pipelines, and plots can be found at [60]

https://mule-tools.gitlab.io/user-library/dilepton/belle

### 4.1  Cross section without cuts

We begin by considering the cross section integrated over all of phase space without any cuts. The relevant $K$ factors are defined as

$$\delta K^{(1)} = \frac{\sigma_{\text{QED}}^{(1)}}{\sigma_{\text{QED}}^{(0)}}\,, \quad \delta K^{(2)} = \frac{\sigma_{\text{QED}}^{(2)}}{\sigma_{\text{QED}}^{(0)} + \sigma_{\text{QED}}^{(1)}} \quad \text{and} \quad \delta K_{\text{EW}} = \frac{\sigma_{\text{EW}}}{\sigma_{\text{QED}}}\,. \tag{34}$$

We further consider the forward-backward asymmetry in the CMS frame which we denote with a $*$

$$A_{\text{FB}}(\sigma) = \frac{\int_0^{\pi/2} \text{d}\theta^*_{\tau^-} \frac{\text{d}\sigma}{\text{d}\theta^*_{\tau^-}} - \int_{\pi/2}^{\pi} \text{d}\theta^*_{\tau^-} \frac{\text{d}\sigma}{\text{d}\theta^*_{\tau^-}}}{\int_0^{\pi/2} \text{d}\theta^*_{\tau^-} \frac{\text{d}\sigma}{\text{d}\theta^*_{\tau^-}} + \int_{\pi/2}^{\pi} \text{d}\theta^*_{\tau^-} \frac{\text{d}\sigma}{\text{d}\theta^*_{\tau^-}}} = \frac{\sigma_{\text{f}} - \sigma_{\text{b}}}{\sigma_{\text{f}} + \sigma_{\text{b}}}. \qquad (35)$$

Following Belle's convention [61], the angles $\theta_{\tau^\pm}$ are defined w.r.t. the incoming positron.[4] At a given order, $A_{\text{FB}}$ is defined to also contain all contributions below it, i.e.

$$A_{\text{FB}}\Big(\sigma^{(\ell)}_{\text{QED}}\Big) \equiv A_{\text{FB}}\Big(\sum_{j=0}^{\ell} \sigma^{(j)}_{\text{QED}}\Big), \quad A_{\text{FB}}\Big(\sigma_{\text{EW}}\Big) \equiv A_{\text{FB}}\Big(\sigma_{\text{QED}} + \sigma_{\text{EW}}\Big). \qquad (36)$$

The cross sections and asymmetries are shown in Table 1 for the unpolarised case and the cases where both electrons are polarised parallel $(+)$ or anti-parallel $(-)$ w.r.t. their direction of flight with a degree of polarisation of 70% in their rest frames. Note that in the QED case, parity invariance implies that there are only two independent configurations since

$$\sigma_{\text{QED}}(++) = \sigma_{\text{QED}}(--) \quad \text{and} \quad \sigma_{\text{QED}}(+-) = \sigma_{\text{QED}}(-+). \qquad (37)$$

In the EW case, parity is violated but CP is still conserved. Hence,

$$\sigma_{\text{EW}}(+-) \neq \sigma_{\text{EW}}(-+) \quad \text{but} \quad \sigma_{\text{EW}}(++) = \sigma_{\text{EW}}(--), \qquad (38)$$

implying three independent configurations.

The angular distributions used for $A_{\text{FB}}$ are shown in Figure 2 for the unpolarised case. We note that even though the NNLO-QED and EW corrections are similar in size at the level of $\text{d}\sigma/\text{d}\theta^*$, the latter are much smaller for the integrated cross section. This is because the NNLO-QED corrections are symmetric while the EW corrections are largely antisymmetric as can be clearly seen in Figure 2. The dominant antisymmetry of the EW corrections is due to the coupling structure. Calculation at tree-level (unpolarised) with $m = M = 0$ shows that the leading symmetric contribution $\sim \cos^2 \theta$ is suppressed by $(V_f/A_f)^2 = (1 - 4s_W^2)^2 \approx 0.009$ compared to the antisymmetric one $\sim \cos \theta$

$$\frac{\text{d}\sigma^{(0)}_{\text{EW}}}{\text{d}\theta^*_{\tau^\pm}} \sim \left( \cos \theta - \frac{V_f^2}{2A_f^2} \cos^2 \theta + \text{const.} + \mathcal{O}\left(\frac{1}{M_Z^2}\right) \right). \qquad (39)$$

As a result, the integrated cross section $\sigma^{(0)}_{\text{EW}}$ is almost eliminated by a cancellation between forward and backward scattering

$$\sigma^{(0)}_{\text{EW}} = \underbrace{\Big((-2.516\,\text{pb})_{\text{f}} + (2.489\,\text{pb})_{\text{b}}\Big)}_{-0.067\,\text{pb}} \text{dim.-six} + \mathcal{O}\Big(\frac{1}{M_Z^4}\Big) \qquad (40)$$

and $\sigma_{\text{EW}}$ is dominated by $\sigma^{(1)}_{\text{EW}}$

$$\sigma^{(1)}_{\text{EW}} = \underbrace{\Big((0.015\,\text{pb})_{\text{f}} + (0.192\,\text{pb})_{\text{b}}\Big)}_{0.207\,\text{pb}} \text{dim.-six} + \mathcal{O}\Big(\frac{1}{M_Z^4}\Big). \qquad (41)$$

The values in (40) and (41) are given for the unpolarised case. If polarisation effects are taken into account $\sigma^{(0)}_{\text{EW}}$ and $\sigma^{(1)}_{\text{EW}}$ are similar in size.

Let us also consider the effects of taking further terms in the $1/M_Z$ expansion. If (40) is extended by dimension-six-squared[5], and dimension-eight contributions, we find

$$\sigma^{(0)}_{\text{EW}} = \underbrace{\Big((-2.516\,\text{pb})_{\text{f}} + (2.489\,\text{pb})_{\text{b}}\Big)}_{-0.067\,\text{pb}} \text{dim.-six} + \underbrace{\Big((0.008\,\text{pb})_{\text{f}} + (0.007\,\text{pb})_{\text{b}}\Big)}_{0.015\,\text{pb}} \text{(dim.-six)}^2$$
$$+ \underbrace{\Big((-0.034\,\text{pb})_{\text{f}} + (0.033\,\text{pb})_{\text{b}}\Big)}_{-0.001\,\text{pb}} \text{dim.-eight} + \mathcal{O}\Big(\frac{1}{M_Z^6}\Big) = -0.053\,\text{pb}. \qquad (42)$$

---

[4]To convert to the more common convention of defining the angle w.r.t. the incoming electron, one would set $\theta \to \pi - \theta$ and $A_{\text{FB}} \to -A_{\text{FB}}$.

[5]This means the contributions where a dimension-six operator was interfered with itself rather than the pure QED amplitude.

| polarisation (00) | | | | |
|---|---|---|---|---|
| | $\sigma_{\mathrm{QED}}^{(0)}$ | $\sigma_{\mathrm{QED}}^{(1)}$ | $\sigma_{\mathrm{QED}}^{(2)}$ | $\sigma_{\mathrm{EW}}$ |
| $\sigma$ / pb | 771.640 | 139.286 | 4.158 | 0.155 |
| $\delta K$ / % | | 18.051 | 0.457 | 0.017 |
| $A_{\mathrm{FB}}$ | 0 | 0.012 | n/a | 0.006 |

| polarisation ($\pm\pm$) | | | | |
|---|---|---|---|---|
| | $\sigma_{\mathrm{QED}}^{(0)}$ | $\sigma_{\mathrm{QED}}^{(1)}$ | $\sigma_{\mathrm{QED}}^{(2)}$ | $\sigma_{\mathrm{EW}}(++)$  $\sigma_{\mathrm{EW}}(--)$ |
| $\sigma$ / pb | 393.537 | 72.922 | 2.537(3) | 0.014 |
| $\delta K$ / % | | 18.530 | 0.544 | 0.003 |
| $A_{\mathrm{FB}}$ | 0 | 0.012 | n/a | 0.006 |

| polarisation ($\mp\pm$) | | | | | |
|---|---|---|---|---|---|
| | $\sigma_{\mathrm{QED}}^{(0)}$ | $\sigma_{\mathrm{QED}}^{(1)}$ | $\sigma_{\mathrm{QED}}^{(2)}$ | $\sigma_{\mathrm{EW}}(-+)$ | $\sigma_{\mathrm{EW}}(+-)$ |
| $\sigma$ / pb | 1149.744 | 205.648 | 5.782(1) | 0.105 | 0.486 |
| $\delta K$ / % | | 17.886 | 0.427 | 0.008 | 0.036 |
| $A_{\mathrm{FB}}$ | 0 | 0.012 | n/a | 0.006 | 0.006 |

Table 1: Cross sections and asymmetries for $e^+e^- \to \tau^+\tau^-$ up to NNLO-QED and NLO-EW for all polarisation configurations ($e^+e^-$). When the electrons are polarised, the degree of polarisation is 70% in their respective rest frames. Unless otherwise indicated, all digits are significant.

From this it is clear that at the differential level the expansion is perfectly justified. However, due to the aforementioned antisymmetry of the dimension-six term, the total LO-EW cross section receives sizeable corrections from the dimension-six-squared term. Hence, to avoid these effects, we include the LO-EW contribution without expansion in $1/M_Z$.

Note that the zero crossing of the EW corrections in Figure 2 does not happen at exactly 90° but is slightly offset due to the small symmetric part in (39).

At LO in QED, $A_{\mathrm{FB}}$ is exactly zero as expected, while at NLO in QED, the mixed tauonic-electronic contribution induces a finite but small asymmetry. This is similar to the NLO-QCD effects resulting in a non-zero $A_{\mathrm{FB}}$ for the hadronic $tt$ production [62]. In principle, this would continue at NNLO in QED. However, the purely electronic contributions considered here are perfectly symmetric w.r.t. $\theta_{\tau\pm}^*$ and therefore do not contribute to $A_{\mathrm{FB}}$. The EW corrections are almost perfectly antisymmetric but much smaller than the QED corrections. This means that the full NNLO-QED corrections will be required to give a meaningful result for $A_{\mathrm{FB}}$.[6] However, unless calculation of the QED two-loop matrix elements with full $m_e$ dependence becomes available, it will not be possible to do this for the ($\pm\pm$)-polarisation configuration which requires a helicity flip that cannot be obtained with massification alone.

## 4.2 Predictions for Belle II

Next, we tailor our calculation to Belle II. The detector is asymmetric since the electron beam's energy is higher than the positron beam's

$$E_{e^-}^{(\mathrm{in})} = 7\,\mathrm{GeV} \quad \text{and} \quad E_{e^+}^{(\mathrm{in})} = 4\,\mathrm{GeV}\,. \tag{43}$$

We approximate the detector's geometric acceptance by requiring that the tau leptons are produced within the geometric acceptance [61]

$$17° < \theta_{\tau\pm} < 150°\,. \tag{44}$$

The angular distribution of the outgoing taus is shown in Figure 3. The LO distribution vanishes below $\approx 53°$ because of the cut on the other particle. However, once real emission is allowed, the angle can become much smaller. Once again, the EW corrections are similar in size to the NNLO-QED ones.

The SuperKEKB beams are currently unpolarised which is reflected in our calculation. However, recent proposals suggest that this could be changed in the future, aiming for 70% polarisation [63]. To study this

---

[6]While this paper was under review, the full NNLO-QED corrections were calculated [25] for $e\mu \to e\mu$, albeit at without polarisation and lower energies where numerical instabilities are less pronounced.

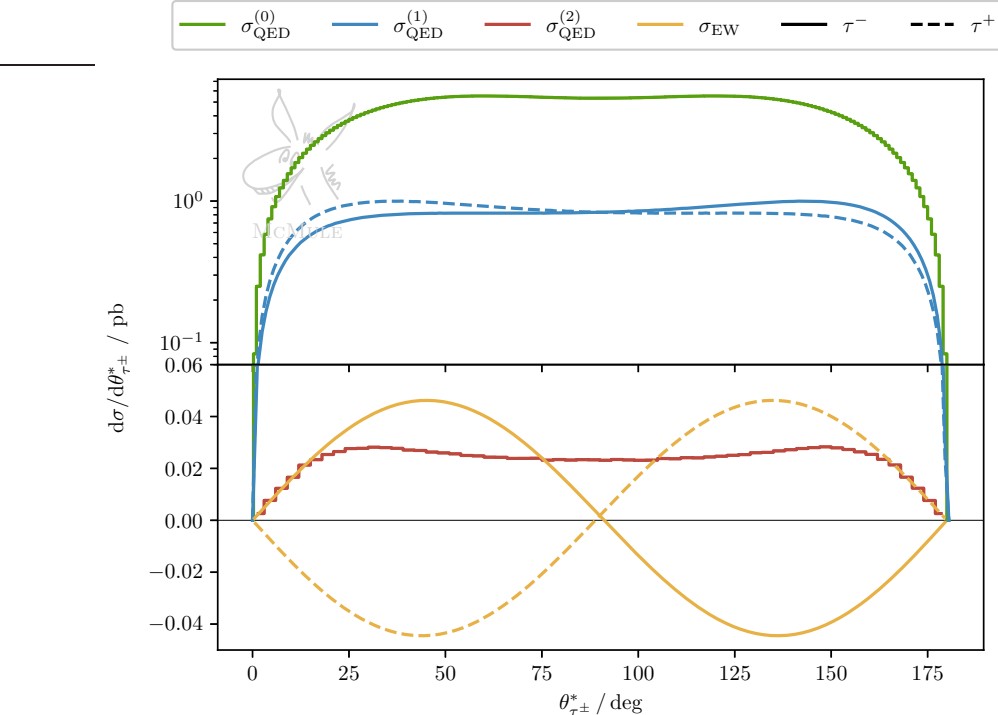

Figure 2: The angular distribution of the two taus in the final state for an unpolarised initial state in the CMS frame. Note that the scale changes from linear to logarithmic at +0.06.

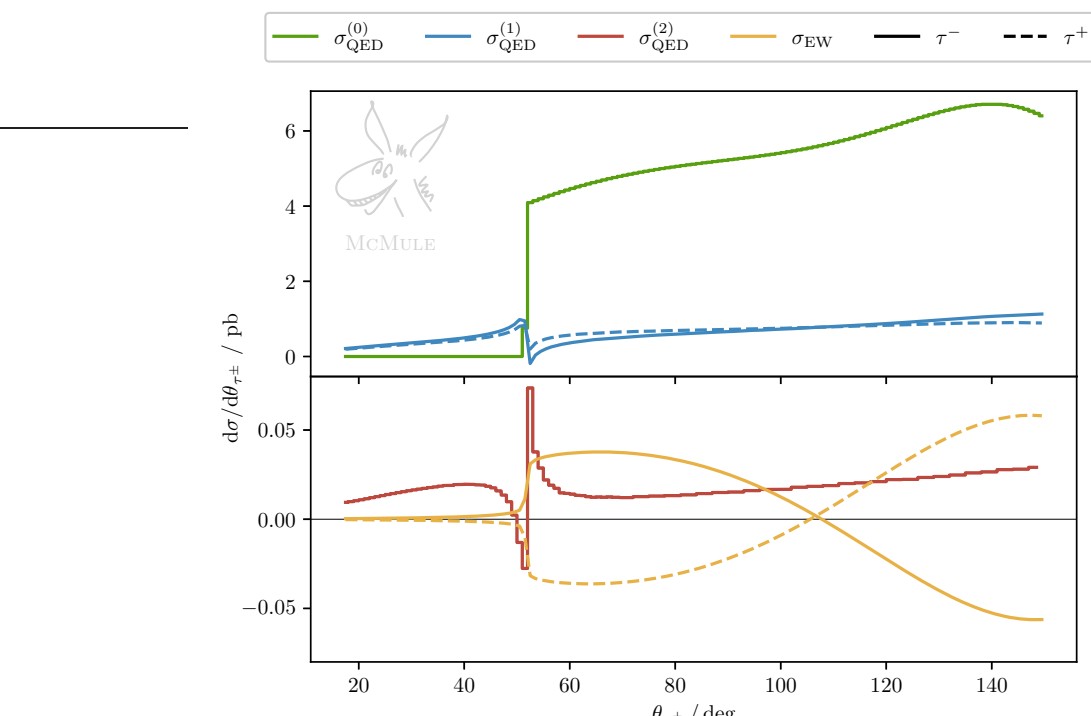

Figure 3: The angular distribution of the two taus in the lab frame for unpolarised initial states. Note that, because we only considered the dominant NNLO-QED corrections, the $\sigma_{\mathrm{QED}}^{(2)}$ curves for $\theta_{\tau^+}$ and $\theta_{\tau^-}$ are identical.

case, we consider the ratio between the polarised and unpolarised angular distributions of the $\tau^-$, both in the lab frame with cuts ($\theta_{\tau^-}$) and the CMS frame without ($\theta_{\tau^-}^*$)

$$\mathcal{R}^{(*)}(\pm+) = \frac{\mathrm{d}\sigma(\pm+)/\mathrm{d}\theta_{\tau^-}^{(*)}}{\mathrm{d}\sigma(00)/\mathrm{d}\theta_{\tau^-}^{(*)}} - \frac{\sigma(\pm+)}{\sigma(00)}. \tag{45}$$

Note that the first term in (45) is not centred around one but instead around $\sigma(\pm+)/\sigma(00)$. Hence, we subtract this overall shift to centre $\mathcal{R}(\pm+)$ around zero to make the comparison between $\mathcal{R}(++)$ and $\mathcal{R}(-+)$ easier.

$\mathcal{R}^*$ and $\mathcal{R}$ are shown in Figure 4. Let us first consider the simpler $\mathcal{R}^*$ (Figure 4a). Since the outgoing taus are not polarised, $\mathcal{R}^* = 0$ at LO. At and beyond NLO, this is no longer true because hard-collinear ISR causes a helicity flip in the emitter.[7] With (45) normalised as it is, adding all polarisations results once again in a flat line, meaning that we have recovered the unpolarised result. Boosting to the lab frame (Figure 4b) stretches the distributions for forward emissions ($53° \lesssim \theta_{\tau^-} \lesssim 120°$) and squeezes them for backward ($120° \lesssim \theta_{\tau^-} \leq 150°$). Further, since the cuts (44) mean that hard emission is required for $\theta_{\tau^-} \lesssim 53°$, the effect is significantly enhanced.

# 5   Conclusion

We have presented a fully differential calculation of the dominant NNLO-QED and NLO-EW corrections for di-lepton productions, including fermionic and bosonic corrections. We find that the EW corrections are of similar size to the NNLO-QED ones for $\sqrt{s} \approx 10.5\,\mathrm{GeV}$ meaning they are vital for Belle II. To perform this calculation, we have extended the strategy of next-to-soft stabilisation to polarised observables, which, combined with OpenLoops, allows for a fast and stable evaluation of the real-virtual matrix element.

All matrix elements were implemented in the parton-level Monte Carlo code McMule which allows the user to calculate arbitrary IR-safe observables. As a first example, we have calculated differential predictions for Belle II, both for polarised and unpolarised initial states.

Since this calculation only includes the dominant contribution, a natural next step would be the inclusion of the full set of NNLO-QED corrections, especially when considering asymmetries such as $A_{\mathrm{FB}}$. The relevant two-loop matrix elements are known in the unpolarised case for $m_e = 0$ [16] but would need to be extended, in a first step, to the polarised case. Finally, this work will need to be combined with [25] to properly include the numerically delicate real-virtual corrections. Work to that end is currently in progress.

Should even higher precision be required, resummation is required. Currently, McMule is calculating strictly at fixed order which means that some important logarithmically-enhanced contributions are not considered beyond NNLO-QED. These can be resummed to all orders using fragmentation functions (for final state) or parton distribution functions (for initial state). For a recent review on this topic, see [64] and references therein. However, this was not done in the present study as any analytic resummation limits what observables can be calculated. Further, there is an effort within the McMule collaboration to include a YFS parton shower that is similar to PHOTONS++ [65] and matched to NNLO-QED.

## Acknowledgement

We are very grateful to our colleagues in the McMule collaboration, esp. Tim Engel, Franziska Hagelstein, Marco Rocco, and Adrian Signer, for supporting the implementation of this process, providing cross-checks, and comments to the manuscript. We would further like to thank Dominik Stöckinger for many fruitful discussions related to EW calculations and effective theories. Next, we are grateful to Fred Jegerlehner for useful discussion regarding the treatment of the EW-HVP corrections. We are also grateful to Max Zoller for providing the real-virtual matrix element in OpenLoops as well as for assisting us with its proper usage. Without the impressive numerical stability of OpenLoops this project would not have been possible. Finally, we would like to thank Peter Richardson and Lois Flower for useful discussions about the polarised distributions. YU acknowledges support by the UK Science and Technology Facilities Council (STFC) under grant ST/T001011/1. SK acknowledges partial support by the Swiss National Science Foundation (SNF) under grant 207386.

---

[7]We verified this with a dedicated run where we required that the initial-state emission is harder than $E_\gamma^* > 0.2\,\mathrm{GeV}$ and $\cos \sphericalangle\left(p_{e^-}^*, p_\gamma^*\right) > 0.8$ in the CMS frame.

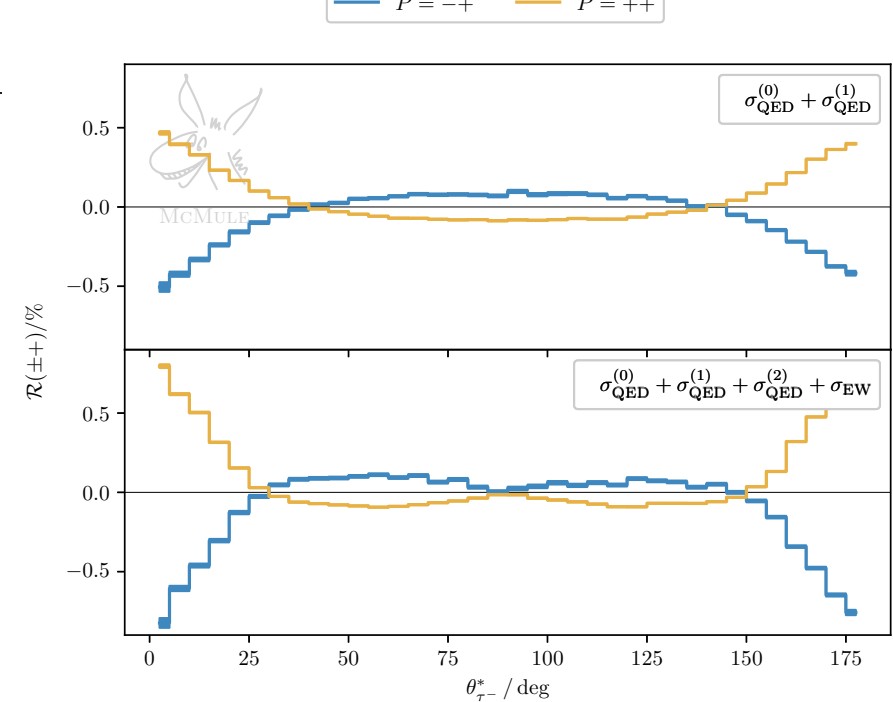

(a) $\mathcal{R}^*$ in the CMS frame without cuts at NLO-QED (upper panel) and for the full result (lower panel).

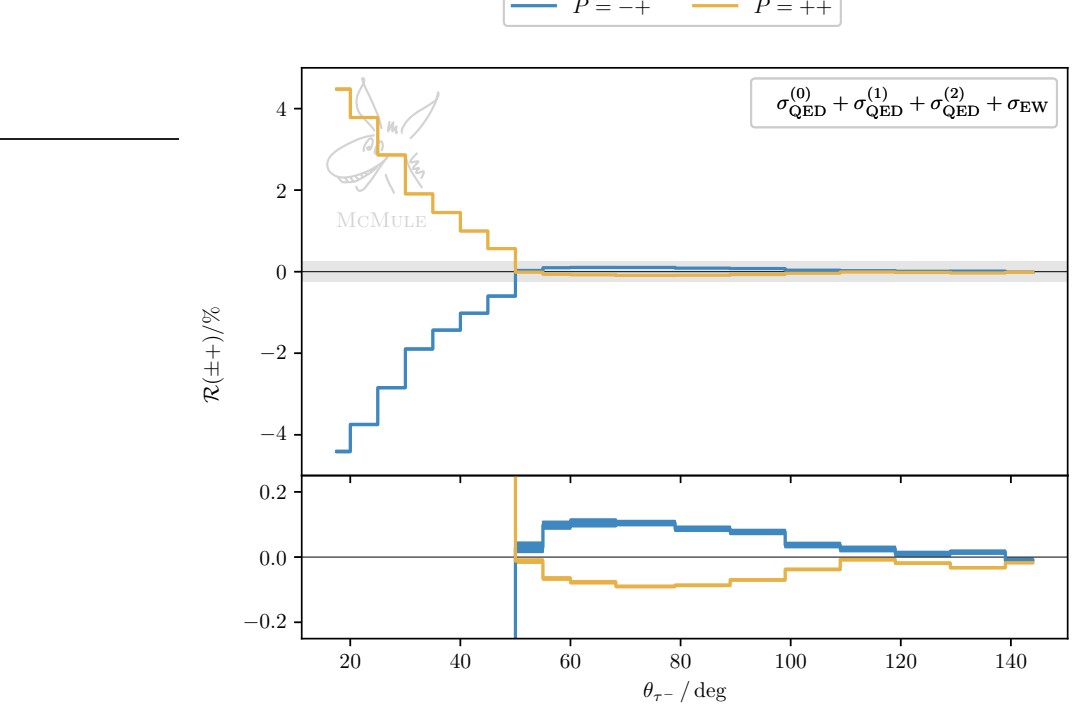

(b) $\mathcal{R}$ in the lab frame with cuts for the full result. Note that the lower panel is zoomed in onto the $\theta_{\tau^-}$ region that is allowed at tree-level.

Figure 4: The ratios $\mathcal{R}^{(*)}(++)$ in orange and $\mathcal{R}^{(*)}(-+)$ in blue, both in the CMS frame and in the lab frame. Note that this observable is very insensitive to EW effects and that the shape of the plots originates from the QED corrections. This means that we only need to show two out of the four different polarisation configurations.

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
