# Peer review of "Lepton pair production at NNLO in QED with EW effects"

_SciPost Physics_

## Round 1 · Referee Report · Anonymous (Referee 1) · 2022-12-21

Strengths

1) The calculation is well motivated in the introduction. 2) The results are implemented in a Monte Carlo code allowing for IR-safe observables. 3) The input and the setup are clearly described. 4) The Monte Carlo code and the numerical results are relevant for the Belle II experiment.

Weaknesses

1) Some ingredients of the calculation are not described. 2) A consistent power counting of the corrections that are included is not provided.

Report

The paper presents a calculation for polarised lepton pair production taking into account the dominant (electronic) NNLO QED corrections and the NLO EW corrections in the energy range of the Belle II experiment. The EW corrections are expanded in $E/M_Z$ , and at NNLO QED only the contributions proportional to $Q_l^2 Q_e^6$ are taken into account. The calculation is based on the massless two-loop matrix elements for ee → ll and the next-to-soft stabilisation method and is implemented in the Monte Carlo framework McMule. This implementation allows to calculate arbitrary IR-safe observables. Some illustrative results are presented that are original and relevant for the Belle II experiment. The paper fulfills most of the general acceptance criteria. Some weaknesses need to be improved before an eventual publication.
I recommend publication of the paper in SciPost Physics, once the items listed below have been fixed.

Requested changes

1) It would be desirable to introduce some sort of power counting for the corrections that are taken into account and those that are not. 2) The renormalisation condition for the photon self-energy in Eq. (10) is irrelevant. This equation should be fulfilled due to gauge invariance. In other words, $Σγγ (0)$ in Eq. (13) is zero. 3) The contribution of the renormalised Z-boson energy in Eq. (11) is the one of the fermion loops, as the title of the section states. Why is there no discussion on the corresponding contributions from the EW boson loops? 4) In Eqs. (21) and (22), there should be a minus sign in front of the second term in the numerator of the term on the right-hand side. 5) The result that the LO and the NLO EW contributions to the cross section are of similar size should be discussed in more detail. Is the LO EW contribution suppressed in some way or is the NLO EW contribution enhanced? Is this a consequence of the suppression of the vector coupling of the charged fermions? 6) In the third line of the last paragraph of section 4, it should read “adding all polarisations results ...”.

  • validity: good
  • significance: high
  • originality: high
  • clarity: ok
  • formatting: excellent
  • grammar: good

Author:  Yannick Ulrich  on 2023-03-02  [id 3421]

(in reply to Report 1 on 2022-12-21)

We would like to thank the referee for their detailed comments. We would like to explicitly point out that we have NOT used the massless calculation for $ee \to \ell\ell$ by Bonciani et al. in this work. We focus purely on the corrections due to the electron line, i.e. contributions that are ~$Q_e^6 Q_\tau^2$. We have stressed this further when discussing the power counting and apologise for any confusion we may have caused.

We now give our reply on the specific points raised by the referee.

  1. We have greatly expanded our discussion of the power counting by describing the different contribution in greater detail, both for the QED part and the EW part.

  2. We agree with the referee that this equation is automatically fulfilled and have added a sentence to that effect. For the sake of completeness, we have not fully removed the equation.

  3. The contributions discussed by the referee are included. However, they are not fermionic in nature and therefore do not belong in Section 2.1. We have instead added a discussion in Section 2.2

  4. This is indeed correct, we thank the referee for spotting this mistake.

  5. We have added a discussion to explain this. The angular distribution at LO is almost completely antisymmetric due to the dominance of the axial-vector coupling. This leads to an unusually small cross section once integrated. This is no longer true at NLO resulting in very large relative corrections.

  6. We have corrected this point.

---

## Round 1 · Referee Report · Anonymous (Referee 2) · 2023-1-16

Report

The authors report the results of the calculation
of the dominant (from electronic line) NNLO QED corrections
and of the NLO electroweak corrections to the process
e+e- -> l+l-, with particular application to tau pair
production at Belle II center of mass energies. The manuscript
considers also the case of polarized incoming beams.
For this particular case the authors extend the formalism of the
next-to-soft stabilization technique, useful to obtain numerically
stable results. All the relevant matrix element have been
implemented in the McMule framework, which is used to obtain
the phenomenological results discussed for Belle II typical running
conditions.
The manuscript contains advancements with respect to the literature
on the theoretical predictions for lepton pair production in
e+e- collisions at low/intermediate energies. The results of the study
and related Monte Carlo code are important for Belle II physics analysis
and pave the way for future upgrades.
The presentation of the results is clear and well written.
Therefore I recommend publication of the manuscript on SciPost.

I only have a minor comment/doubt on Eq. (34), where the definition
of FB asymmetry is given by integrating the lepton angle in
the reference frame with the incoming positron along the z-axis,
if I understand correctly. Unless there is a particular convention
in Belle II, to the best of my knowledge, the lepton scattering angle
is typically referred to the incoming lepton beam in e+e- colliders.
  • validity: -
  • significance: -
  • originality: -
  • clarity: -
  • formatting: -
  • grammar: -

Author:  Yannick Ulrich  on 2023-03-02  [id 3422]

(in reply to Report 2 on 2023-01-16)

We would like to thank the referee for their very positive feedback. Regarding the comment they raise about the definition of the lepton scattering angle, we have indeed followed the definition used by Belle where the angle is defined w.r.t. the positron beam. We have stressed this and added a footnote about the (trivial) conversion to the 'conventional' definition of $\theta$.

---

## Round 1 · Referee Report · Anonymous (Referee 3) · 2023-1-18

Strengths

Development of a framework for the simulation of high-precision processes.

Weaknesses

Unclear classification of the hierarchy of the different radiative corrections effect.

Report

The paper “Lepton pair production at NNLO in QED with EW effects” discusses an important topic, namely the precise prediction of the differential cross section for the production of a pair of tau leptons, at BELLE energies. This is fundamental in view of a correct theoretical interpretation of the very precise experimental measurements.

The logic of the paper is clear, the quality of the numerical results presented is good and my overall impression is that it is an interesting contribution. I find unclear the description of the different approximations adopted in the study and the arguments used to argue that they represent an accurate estimate of a complete two-loop calculation.

I would recommend the publication, after the points described below are clarified.

Requested changes

1) Basic definitions What is the definition of EW effects? At LO? At NLO? Is a 1-loop box with a photon and a Z boson between initial and final state considered as QED, EW correction, or completely discarded?

2) QED Can the author precisely state the content of the different orders in QED and the hierarchy of the different contributions? Is NLO-QED a full calculation with ISR, FSR and IFI? At NNLO-QED, do the authors consider only ISR ? Is the renormalization at NNLO-QED level performed?

Are QED corrections to the final state tau applied? Are they precise enough for the goals of this study, e.g. for the precise prediction of the asymmetries?

Does the claim about the relevance of electronic corrections mean that all the logarithms of the electron mass are included? What about the powers of the electron mass? Why should they be more relevant than powers of the tau lepton mass? What is the exact improvement of this study, compared to reference [16] ?

Do the two-loop self-energy photon vacuum polarization corrections with one electron loop belong to the set studied here (they fulfill the counting of charges described in the introduction, but they are not described anywhere) ?

3) EW The exact NLO-EW corrections are expanded. It is not clear what is the expansion parameter: it should be an adimensional quantity, rather than 1/MZ^2; it is different if the ratio is me^2/MZ^2 or s/MZ^2. Given the role of the asymmetries, how are the radiative corrections to the Z boson couplings to fermions handled? How is the electric charge renormalized? And how the weak mixing angle?

The authors mention the choice of alpha, Gmu, MZ as input parameters. Is the parameter Delta r evaluated in this calculation? If yes, is it expanded in the same way as the rest of the EW corrections?

Why the title “Bosonic corrections” for Section 2.2, given the content which is about the structure of a two-loop calculation, with some technicalities?

The choice about the missing splitting of the EW part in sigma^{(0)}{EW} + sigma^{(1)} is not clear. What is the definition of sigma^{(0)}_{EW}? Is it gauge invariant? If so, why is it acceptable to have quantum corrections larger than the previous order? Is there any physical reason, perhaps a cancellation at LO or an enhancement at NLO ?

4) Phenomenology Can the authors explain the perfectly antisymmetric behavior of the EW corrections in Figure 2? Does it depend on their expansion recipe and definition of EW correction? At the end of Section 4.1, what is the “perfectly asymmetric” behavior of the EW corrections? Do they mean antisymmetric?

The paper presents a fixed-order calculation of initial state QED corrections. Large mass logarithms are generated by these emissions. Given the high-precision goal of the study, how does this approach compare with a radiator, structure function, of lepton PDFs approach, where the all-order resummation of such large log corrections is performed ?

—————- I would like to ask the authors to answer the questions listed above, before recommending the paper for publication.

  • validity: ok
  • significance: good
  • originality: ok
  • clarity: low
  • formatting: good
  • grammar: good

Author:  Yannick Ulrich  on 2023-03-02  [id 3423]

(in reply to Report 3 on 2023-01-18)

We would like to thank the referee for their positive feedback and detailed comments, allowing us to clarify our paper. We have added a more detailed discussion of the power counting to clarify what was approximated and how.

We now give our reply on the specific points raised by the referee.

  1. We define EW effects to include all Standard Model effects that are not covered by QED. This means that we include Z and Higgs exchanges at LO and all one-loop and real emission diagrams at NLO. Naturally, this also includes the one-loop box with a photon and a Z boson as an EW corrections as discussed in Section 2.2.

2.1. We have made the definition of the different QED contributions more explicit and explained the hierarchy better. At NLO we indeed include ISR, FSR, and IFI. At NNLO we only consider ISR. Of course, we have performed a full renormalisation to the level that we have calculated. Comments to this effect have been added.

2.2. QED corrections to the final states are included only at NLO. Inclusion of FSR effects at NNLO-QED is trivial but since $Q^2~m_\tau^2$ they are expected to not contribute significantly. This has been explicitly confirmed in the context of mu-e scattering in the new ref. [25]. The inclusion of mixed IFI contribution at NNLO is indeed relevant especially for the asymmetry. We have clarified this shortcoming further.

2.3. Our calculation is exact in the electron mass, i.e. we include all logarithms and power-suppressed terms in everything we calculate. That being said, the IFI contribution will contain terms $\propto \alpha^2 \log(m_e^2/Q^2)$ that we have not calculated. Since the electron mass is much smaller than all other scales, power-suppressed terms are not overly relevant. Powers of the tau mass are more relevant and fall under IFI and FSR that are expected to be smaller than the ISR contributions.

Reference [16] provides the two-loop unpolarised matrix element for the IFI contribution with vanishing electron mass. While a very impressive calculation, it is a divergent quantity until combined with real corrections. This was done in the new [25] for the case of $\mu-e$ scattering.

We have clarified these points in the manuscript.

2.4. The two-loop self-energy photon vacuum polarisation corrections are indeed included. We have added a sentence and a reference at the end of Section 2.1 to describe them and apologise for the confusion caused.

3.1. We have added a discussion of how the expansion is performed. In essence, we consider all ratios of light scales ($s$, $t$, $m_e^2$, $m_\tau^2$) to heavy scales ($M_Z^2$, $M_W^2$, $M_H^2$) to be equally small and expand in all of them. For simplicity, we write this as an expansion in $1/M_Z$, similar to how one expands in an effective field theory in $1/\Lambda$.

3.2. As discussed in Section 2, the electric charge and weak mixing angle are both both renormalised in the on-shell scheme. We are not exactly sure what the referee is refering to with this question. However, the radiative corrections to the Z boson coupling (i.e. triangle diagrams) are also calculated in an expansion of ${m^2,s}/M_Z^2$ and then renormalised in the on-shell sheme.

3.3. We treat $\Delta r$ as mechanism to obtain the input parameters in the scheme of our choice and do not expand it. This has now been explained after (3).

3.4. The title of Section 2.2 was chosen in contrast to Section 2.1 which covers fermionic loops. Since bosonic corrections includes photonic ones it justifies the discussion of the two-loop calculation. However, we have now also used this section to discuss the point raised in Question 1.

3.5. $\sigma_{EW}^{(0)}$ ($\sigma_{EW}^{(1)}$) includes all contributions that have at least one heavy particle (Z, W, Higgs) at LO (NLO). This makes the split gauge invariant. However, neither $\sigma_{EW}^{(0)}$ nor $\sigma_{EW}^{(1)}$ are physical in their own right since QED needs to be included. We have now extended this discussion to include why the integrated cross section $\sigma_{EW}^{(0)}$ is so small compared to $\sigma_{EW}^{(1)}$.

4.1. We have added a discussion to explain the antisymmetric behaviour in Figure 2 which is due to ratio of vector-to-axial coupling $(V_f/A_f)^2 = (1-4s_W^2)^2 \approx 0.009$. This changes slightly when considering higher-order terms, either in the expansion or for radiative corrections, but is independent of the definition of the EW corrections.

4.2. We indeed meant antisymmetric and thank the referee for pointing this out.

4.3. While we agree that these logarithms can be predicted and eventually resummed using other tools, the present study is only considering fixed-order results and methods and hence does not include a comparison with any all-order resummation. Nevertheless, we have pointed out these techniques in our conclusion, noting that these calculations need to performed to reach higher precisions.

---

## Editorial Decision

resubmitted